# Within-trial cost-effectiveness of lifestyle intervention using a 3-tier shared care approach for pregnancy outcomes in Chinese women with gestational diabetes

Weiqin Li[1,2,3], Cuiping Zhang[1], Junhong Leng[1], Ping Shao[1], Huiguang Tian[1], Fuxia Zhang[1], Ling Dong[1], Zhijie Yu[4], Juliana C. N. Chan[5], Gang Hu[6], Ping Zhang[7‡*], Xilin Yang[8‡*]

1 Tianjin Women and Children's Health Centre, Tianjin, China, 2 Department of Epidemiology, Capital Institute of Pediatrics, Beijing, China, 3 Graduate School of Peking Union Medical College, Beijing, China, 4 Population Cancer Research Program and Department of Pediatrics, Dalhousie University, Halifax, Canada, 5 Department of Medicine and Therapeutics, Hong Kong Institute of Diabetes and Obesity and The Chinese University of Hong Kong-Prince of Wales Hospital-International Diabetes Federation Centre of Education, Hong Kong SAR, China, 6 Chronic Disease Epidemiology Laboratory, Pennington Biomedical Research Center, Baton Rouge, Louisiana, United States of America, 7 Division of Diabetes Translation, Centers for Disease Control and Prevention, Atlanta, Georgia, United States of America, 8 Department of Epidemiology and Biostatistics, School of Public Health, Tianjin Medical University, Tianjin, China

☉ These authors contributed equally to this work.
‡ PZ and XY also contributed equally to this work.
* pzhang@cdc.gov (PZ); yangxilin@tmu.edu.cn (XY)

**Data Availability Statement:** All relevant data are within the paper and its Supporting Information files.

## Abstract

This study assessed within-trial cost-effectiveness of a shared care program (SC, n = 339) for pregnancy outcomes compared to usual care (UC, n = 361), as implemented in a randomized trial of Chinese women with gestational diabetes (GDM). SC consisted of an individualized dietary advice and physical activity counseling program. The UC was a one-time group education program. The effectiveness was measured by number needed to treat (NNT) to prevent one macrosomia/large for gestational age (LGA) infant. The cost-effectiveness was measured by incremental cost-effectiveness ratio in terms of cost (2012 Chinese Yuan/US dollar) per case of macrosomia and LGA prevented. The study took both a health care system and a societal perspective. This study found that the NNT was 16/14 for macrosomia/LGA. The incremental cost for treating a pregnant woman was ¥1,877 ($298) from a health care system perspective and ¥2,056 ($327) from a societal perspective. The cost of preventing a case of macrosomia/LGA from the two corresponding perspectives were ¥30,032/¥26,278 ($4,775/$4,178) and ¥32,896/¥28,784 ($5,230/$4,577), respectively. Considering the potential severe adverse health and economic consequences of a macrosomia/LGA infant, our findings suggest that implementing this lifestyle intervention for women with GDM is an efficient use of health care resources.

**Funding:** This project was supported by the National Key Research and Development Program of China (Grants No: 2018YFC1313900, 2018YFC1313903, 2016YFC1300101 and 2016YFC0900602), and BRIDGES (Project No: LT09-227). BRIDGES is an International Diabetes Federation program supported by an educational grant from Lilly Diabetes. The funders had no role in study design, data collection and analysis, decision to publish, or preparation of the manuscript.

**Competing interests:** Lilly Diabetes is a commercial source, but this does not alter our adherence to PLOS ONE policies on sharing data and materials.

## Introduction

Gestational diabetes mellitus (GDM), defined as diagnosis of hyperglycemia of any degree during pregnancy, is associated with adverse pregnancy outcomes including macrosomia [1], preterm birth [1, 2], shoulder dystocia, birth trauma and neonatal morbidities [3–5]. GDM also predisposes offspring born to women who had GDM during the index pregnancy to higher risk of childhood obesity [6], and the mothers themselves to higher risk of developing diabetes during their lifetime [7, 8].

Lifestyle intervention is effective in improving pregnancy outcomes among women with GDM [9, 10]. The Australian Carbohydrate Intolerance Study in Pregnant Women (ACHOIS) reported that intensive lifestyle intervention reduced the rate of serious perinatal complications, defined as death, shoulder dystocia, bone fracture, and nerve palsy, macrosomia and preeclampsia among women with GDM [9]. GDM in this study was defined by the World Health Organization's criteria (fasting plasma glucose levels ≥7.8 mmol/l, or 2-hour plasma glucose levels between 7.8 mmol/l and 11.1 mmol/l) [11], based on a 75-gram 2-hour oral glucose tolerance test (OGTT). Another multicenter randomized trial reported that intensive lifestyle intervention reduced the rates of macrosomia [10], shoulder dystocia and pregnancy-induced hypertension (PIH) [10] in women with mild GDM, defined by the Fourth International Workshop-Conference on GDM's criteria (a fasting glucose level < 5.3 mmol/l and one glucose measurement that exceeds the following thresholds: 1-hour, 10.0 mmol/l; 2-hour, 8.6 mmol/l; and 3-hour, 7.8 mmol/l) [12], based on a 100-gram 3-hour OGTT.

We conducted a randomized trial of Chinese women with GDM using the International Association of Diabetes and Pregnancy Study Group's (IADPSG) criteria (fasting plasma glucose levels ≥5.1 mmol/l, or 1-hour plasma glucose levels ≥10.0 mmol/l, or 2-hour plasma glucose levels ≥8.5 mmol/l, based on a 75-gram 2-hour OGTT) [13]. The primary objective of the study was to test if lifestyle intervention delivered in the 3-tier prenatal care system (SC), as compared with usual antenatal care (UC), could improve pregnancy outcomes among Chinese women with GDM. The secondary objective was to assess the cost-effectiveness of the lifestyle intervention program if the program was effective. We previously reported that the lifestyle intervention program reduced the risk of macrosomia and large for gestational age (LGA) [14]. Here we report the results of the cost-effectiveness of the intervention as implemented in the trial.

## Materials and methods

### Brief description of the trial

Details of the trial including background, methods, study outcomes and research settings were described elsewhere [14]. Briefly, between December 2010 and October 2012, 19,847 pregnant women who were between 24 and 28 weeks of gestation underwent a 50-gram 1-hour glucose challenge test (GCT) at primary hospitals within the 3-tier's health care network in the six central urban districts of Tianjin, China. The 3-tier network prenatal care system consists of more than 300 primary hospitals throughout the city (first level), 16 district-level Women and Children's Health Centers (WCHC), more than 50 secondary obstetric hospitals (second level), 1 city-level WCHC (TWCHC) and 7 tertiary obstetric hospitals (third level). Structured prenatal care was delivered within the network. First, all pregnant women in the city were registered at a primary hospital and received their routine prenatal care until the 32nd gestational week. Then they were referred to one of the secondary or tertiary obstetric hospitals for the remainder of their prenatal care. If women experienced pregnancy complications, they were referred to third-level treatment facilities.

The GCT test identified 2921 women with plasma glucose (PG) values of $\geq$7.8 mmol/L. A following 75-gram OGTT test was performed in these women and 1,440 of them were diagnosed with GDM using the IADPSG's criteria [14]. Of the 1440 women with GDM, the 948 who agreed to participate in the study and did not meet any of the exclusion criteria were randomly assigned to either the shared care (SC) group (n = 474) or the usual care (UC) group (n = 474). However, the intervention effect on 242 participants might have been contaminated due to (1) unavailability of separate spaces for the intervention and control groups at the central intervention site due to building renovation; and (2) the staff at this study site did not follow data collection procedures as specified in the study protocol (i.e., the same staff members collected data for both study groups due to insufficient manpower). Therefore, we excluded those 242 participants from the main analysis [14]. In addition, six study participants who chose to deliver their infants at hospitals outside Tianjin were excluded. Thus, 700 study participants were included in the final analysis: 339 from the SC and 361 from the UC. A simple randomization procedure without replacement (i.e., by the time sequence of visits to the clinic and a list of priori computer-generated random assignment status by X.Y.) was used to perform the random assignment by the intervention team members [14]. The study flow diagram (S1 Fig) shows the details. Major demographic and clinical information of the study participants are listed in S1 Table. There were no significant differences between the two groups for the baseline clinical and biochemical characteristics. A more detailed description of the study participants can be found in our previous study [14]. The RCT was registered at Clinicaltrials. gov (NCT01565564). Ethics approval of the original study was granted by the Human Subjects Committee of the Tianjin Women's and Children's Health Center, and all participants provided written informed consent.

Women in the UC group were offered one hospital-based group diabetes education session lasting for 30–40 minutes with a focus on diet and physical activity. In addition to UC, the SC group was offered structured intensive lifestyle intervention, including (1) one group general diabetes/GDM counseling and education materials at enrollment, (2) two detailed individualized consultations from trained doctors and laboratory testing at the time of the 30[th] and 34[th] gestational weeks, (3) three health education sessions with trained nurses at the time of the 27[th], 29[th] and 33[th] gestational weeks, and (4) regular self-monitoring of blood glucose (SMBG) [14]. The dietary advice and physical activity counseling included recommendations on a tailored diet and a minimum 30 minutes daily of light/moderate physical activity such as walking. The diet recommendation was based on pre-pregnancy body mass index (BMI), i.e., 35 kcal per day for one kilogram of body weight (35 kcal/day/kg) for women with BMI at <18.5 kg/m$^2$; 30–35 kcal/day/kg for women with BMI at 18.5–23.9 kg/m$^2$; 25–30 kcal/day/kg for women with BMI at 24.0–27.9 kg/m$^2$; 25 kcal/day/kg for women with BMI at 28.0 kg/m$^2$ and more. In addition, the SC group was offered a free glucose meter with a memory function and free tests trips. The SC participants were also asked to monitor blood glucose (SMBG) 4 times a day for the initial two weeks and then daily at different time points (alternating between pre-breakfast and 2 hours after each of the three meals) until the end of pregnancy. Glycemic control goals were set at $\geq$3.5-$\leq$5.1 mmol/l for fasting capillary blood glucose and $\leq$7.0 mmol/l for 2-hour postprandial capillary blood glucose up to the 36[th] gestational week and $\leq$8.0 mmol/l from the 36[th] week onwards. If blood glucose values exceeded the target values two times or more during a 2-week interval or the 2-hour postprandial capillary blood glucose exceeded 9.0 mmol/l once during a 1-week period, insulin therapy would be recommended by the intervention team and started by a senior obstetrician. The obstetric care was the same for both study groups.

## Cost-effectiveness analysis

We measured the cost-effectiveness of this lifestyle intervention using incremental cost-effectiveness ratio (ICER), which was calculated as the incremental cost divided by incremental effectiveness. The incremental cost is the net cost between the SC and UC, and the incremental effectiveness was the net effectiveness between the two study groups. The study took two perspectives: a health system perspective, which considered the direct medical cost only; and a societal perspective, which considered direct medical cost, direct non-medical cost and indirect cost. As we performed a within-trial analysis, our analytical time horizon was from randomization to postnatal hospital discharge of the mother and baby. Analyses were performed on an intention to treat basis. All costs were reported in Chinese yuan (CNY, ¥) (1 CNY = 0.159 US dollars, 31 December 2012).

## Effectiveness of the intervention

We measured the effectiveness of the intervention using the number needed to treat (NNT) for two health outcomes: macrosomia and LGA. Macrosomia was defined as birth weight ≥4,000 gram. LGA was defined by birth weight above the gender- and gestational age- specific 90th percentile using the Tianjin local references. NNT was calculated as the inverse of the absolute risk reduction of macrosomia/LGA.

## Direct medical costs

Direct medical costs included the intervention costs and the routine surveillance, obstetric and neonatal complications attributable to GDM. The costs associated with various glucose tests to identify women with GDM were the same for both the SC and UC groups. These tests included a GCT at primary hospitals, an OGTT between the 24th and 28th gestational weeks. The cost associated with the initial hospital-based group diabetes education session upon diagnosis of GDM was also the same for both groups. Additional costs for the SC group included (1) one group general diabetes/GDM counseling and education materials at enrollment, (2) two detailed individualized consultations from trained doctors and laboratory testing at the time of the 30th and 34th gestational weeks, (3) three health education sessions with trained nurses at the time of the 27th, 29th and 33th gestational weeks, and (4) SMBG. All women in the SC group were offered a free glucose meter with a memory function and free test strips. Similar to the SC women, additional costs for the UC group included costs associated with a simple diabetes/GDM consultation and laboratory testing at the 34th gestational. The specific cost items included are listed in Table 1. Itemized cost information for each study participant was recorded during the trial period.

Costs for routine surveillance, obstetric and neonatal complications attributable to GDM were collected from the electronic database *Pregnant Women Health Records* and also from a questionnaire survey at 4–6 weeks after delivery, regardless of the random group assignment. Once women were diagnosed with GDM, they received a hyperglycemia review one week later. Women with GDM were automatically categorized as having a high-risk pregnancy which triggered multidisciplinary clinical management. The management involved clinical review, fetal surveillance and medications if needed at primary hospitals before the 32nd gestational week, and referral to secondary or tertiary hospitals thereafter for continued care until delivery. Both costs of hospitalizations before and during delivery for obstetric and neonatal complications attributable to GDM were included in the analysis. Obstetric complications included poor metabolic control, pre-eclampsia, fetal macrosomia or fetal growth retardation, labor-induction and caesarean delivery. Neonatal complications included hypoglycemia, respiratory distress and other complications, and admission to the neonatal nursery (S2 Table).

**Table 1. Intervention costs of a lifestyle intervention program for women with gestational diabetes in China, per study participant, by study arm.**

| Intervention phase | Shared care (n = 339, CNY, ¥) | Usual care (n = 361 CNY, ¥,) | Incremental cost (CNY, ¥,) | Data Sources |
|---|---|---|---|---|
| *Screening and diagnosis: 24-28th gestational week* | | | | |
| GCT | 15 | 15 | 0 | MI |
| OGTT | 120 | 120 | 0 | MI |
| Plasma insulin | 100 | 100 | 0 | IDF |
| *Hospital-based education session upon diagnosis of GDM* | 24 | 24 | 0 | MI |
| *Enrollment and first intervention* | | | | |
| Group counseling | 50 | 0 | 50 | IDF |
| Education materials | 10 | 0 | 10 | IDF |
| *Individual consultations: 30th and 34th gestational week* | | | | |
| Antenatal visit to Obstetricians | 22 | 9 | 13 | IDF |
| Laboratory testing during 30th gestational week | 19 | 0 | 19 | IDF |
| Laboratory testing during 34th gestational week | 126 | 98 | 28 | IDF |
| Nutrition analysis and diet consultation | 73 | 0 | 73 | IDF |
| Individual consultation | 73 | 12 | 61 | IDF |
| *Health education session: 27th, 29th and 33rd gestational week* | 120 | 0 | 120 | IDF |
| *Regular self-monitoring of blood glucose:* | | | | |
| Glucose meters | 380 | 0 | 380 | IDF |
| Glucose test strips | 200 | 0 | 200 | IDF |
| *Insulin therapy in pregnancy* | 12 | 3 | 9 | IDF |
| **Total** | **1,344** | **381** | **963** | **MI or IDF** |

All costs were reported in Chinese yuan (CNY, ¥) (1 CNY: 0.159 USD, 31 December 2012).

MI, Maternity insurance. IDF, International Diabetes Federation.

## Direct non-medical costs and indirect costs

The cost items included in this category were: the travel expenses and time costs related to the use of health services; food costs associated with dietary changes; time lost due to morbidity and outpatient treatments; and overhead charges such as rental, utilities and maintenance costs for the office space. These costs were obtained via a questionnaire at 4–6 weeks after delivery, as listed in Table 2.

**Table 2. Direct non-medical and indirect costs associated with a lifestyle intervention program for women with gestational diabetes in China, per study participant, by study arm.**

| | Shared care (n = 339, CNY, ¥) | Usual care (n = 361 CNY, ¥,) | Incremental cost (CNY, ¥,) | Sources |
|---|---|---|---|---|
| Travel to follow-up visits | 305 | 283 | 22 | IP |
| Costs of lower glycemic index snacks | 1,095 | 924 | 171 | IP |
| Time lost due to outpatient services | 1,219 | 1,130 | 89 | IP |
| Time lost due to morbidity | 425 | 598 | -173 | IP |
| Overhead charges | 140 | 70 | 70 | IDF |
| **Total** | **3,184** | **3,005** | **179** | **IP or IDF** |

All costs were reported in Chinese yuan (CNY, ¥) (1 CNY: 0.159 USD, 31 December 2012).

IP, Individual payment. IDF, International Diabetes Federation.

### Sensitivity analysis

The first sensitivity analysis was to examine how cost-effectiveness of the intervention would change by excluding data for the 242 women (130 in SC and 112 in UC) for whom the intervention effect could be contaminated as described earlier. The second was to examine how the result would change if the cost of implementing the intervention was higher in different settings by doubling intervention cost.

## Results

SC significantly reduced the rates of macrosomia [SC group vs. UC group: 11.2% (38/339) vs. 17.5% (63/361), p = 0.019] and LGA [SC group vs. UC group: 13.0% (44/339) vs. 19.9% (72/361), p = 0.013] in comparison to UC. The NNT for the intervention was 16 to prevent one macrosomia, and 14 to prevent one LGA.

The estimated cost of various cost components per study participant by study group are described in Tables 1–3 and S2 Table, including intervention cost in Table 1, the non-medical and indirect costs in Table 2, the total cost in Table 3, and costs associated with routine surveillance, obstetric and neonatal complications in S2 Table. The direct medical costs per study participant in the SC and UC groups were ¥10,892 and ¥9,015, respectively, which included intervention costs (¥1,344 and ¥381) and costs of routine surveillance, obstetric and neonatal complications attributable to GDM (¥9,549 and ¥8,634). The direct non-medical and indirect costs per subject were ¥3,187 and ¥3,005 in the SC and UC groups, respectively.

From the health system perspective, the total cost per study participant was ¥10,892 in the SC group and ¥9,015 in the UC group, and the incremental cost was ¥1,877. From the societal perspective, the cost per study participant in the SC/UC groups was ¥14,076/¥12,020, and the incremental cost was ¥2,056 (Table 3). Thus, from the health system perspective, the cost to prevent a case of macrosomia/LGA was ¥30,032/¥26,278, calculated by multiplying NNT by the incremental cost. From the societal perspective, the corresponding cost-effectiveness ratio was ¥32,896/¥28,784 per prevented case of macrosomia/LGA (Table 4).

In the sensitivity analysis, the NNT to prevent one macrosomia/LGA was 32/20. The incremental cost was ¥1,701 and the ICER was ¥54,432/¥34,020 for macrosomia/LGA from the health system perspective. The corresponding estimates from the societal perspective were ¥1,925 for the incremental cost and ¥61,600/¥38,500 for the ICER (Table 4). Doubling the intervention led to a higher incremental cost and ICERs (Table 4).

**Table 3. Total costs of a lifestyle intervention program for women with gestational diabetes in China, per study participant, by study arm.**

|  | Shared care (n = 339, CNY, ¥) | Usual care (n = 361 CNY, ¥,) | Incremental cost (CNY, ¥,) |
|---|---|---|---|
| *Direct medical costs per subject* |  |  |  |
| Intervention costs | 1,344 | 381 | 963 |
| Routine surveillance, obstetric and neonatal complications attributable to GDM | 9,549 | 8,634 | 915 |
| Subtotal | 10,892 | 9,015 | 1,877 |
| *Direct non-medical and indirect costs per subject* | 3,184 | 3,005 | 179 |
| Total cost per subject |  |  |  |
| Health system perspective* | 10,892 | 9,015 | 1,877 |
| Social perspective* | 14,076 | 12,020 | 2,056 |

All costs were reported in Chinese yuan (CNY, ¥) (1 CNY: 0.159 USD, 31 December 2012).

* Health system perspective cost only included the direct medical costs; societal perspective cost included direct medical cost, direct non-medical cost, and indirect cost.

**Table 4. Effectiveness and cost-effectiveness of a lifestyle intervention program for women with gestational diabetes in China.**

| | Effectiveness of intervention | | Incremental cost (CNY, ¥) | | Incremental effectiveness, Macrosomia /LGA (NNT)‡ | Incremental cost-effectiveness ratio (CNY, ¥) | | | |
|---|---|---|---|---|---|---|---|---|---|
| | | | | | | Health care system | | Society | |
| | Macrosomia Incidence (%) | LGA† Incidence (%) | Health care system | Society | | Macrosomia | LGA† | Macrosomia | LGA† |
| **Main analysis** | | | | | | | | | |
| Shared care | 11.2 | 13.0 | 1,877 | 2,056 | 16/14 | 30,032 | 26,278 | 32,896 | 28,784 |
| Usual care | 17.5 | 19.9 | - - | - - | - - | - - | - - | - - | - - |
| **Sensitivity analysis** | | | | | | | | | |
| *Including all 936 women* | | | | | | | | | |
| Shared care | 12.5 | 14.0 | 1,701 | 1,925 | 32/20 | 54,432 | 34,020 | *61,600* | *38,500* |
| Usual care | 15.6 | 18.9 | - - | - - | - - | - - | - - | - - | - - |
| *Doubling the intervention cost* | | | | | | | | | |
| Shared care | 11.2 | 13.0 | 2,840 | 3,019 | 16/14 | 45,440 | 39,760 | 48,304 | 42,266 |
| Usual care | 17.5 | 19.9 | - - | - - | - - | - - | - - | - - | - - |

All costs were reported in Chinese yuan (CNY, ¥) (1 CNY: 0.159 USD, 31 December 2012).

‡ NNT, Number needed to treat to prevent one case.

† LGA, Large for gestational age was defined by gender and gestational age-specific 90th percentiles.

## Discussion

Intensive lifestyle intervention among women with GDM is effective in improving pregnancy outcomes in China [14], but whether this intervention is cost-effective is unclear. Our study results demonstrate that such intervention costs less than ¥33,000 ($5,247) to prevent one macrosomia or LGA. Whether or not implementing this intervention among all women with GDM in China is an efficient use of health care resources depends on the future health and economic consequences of preventing a case of macrosomia or LGA and ultimately society's willingness to pay for avoiding such adverse outcomes.

Spending less than ¥33,000 ($5,247) to prevent a macrosomia/LGA infant seems to represent a good value in terms of efficient use of limited health care resources in China or elsewhere in the world. First, macrosomia/LGA has both short-term and long-term adverse health complications. The immediate adverse health effects include preterm birth, higher rates of postpartum hemorrhage, increased risk of caesarean delivery, birth trauma, as well as low 1-min Apgar scores [15, 16]. Macrosomia/LGA also has long-term adverse health risks for the offspring. A prospective study using a population-based sample of 21,315 mother-child pairs in China examined the risk factors and long-term health consequences of macrosomia, and found that macrosomic infants showed an increased susceptibility to being overweight and/or obesity in childhood [17]. Obesity among children is a significant risk factor for the development of insulin resistance in a dose-response manner [18]. Moreover, macrosomia is an independent risk factor for developing metabolic syndrome in childhood, and even type 2 diabetes [19] and hypertension [20] later in life.

The long-term adverse health outcomes associated with macrosomia/LGA lead to high future health costs. A literature review reported that a child with obesity bears an extra lifetime medical cost of $19,000 as compared to a child of normal weight in the USA [21]. Persons with diabetes have substantially higher lifetime medical expenditures. Zhuo X. et al showed that in the United States the excess lifetime medical costs for people with diabetes diagnosed at ages 40, 50, 60, and 65 years were $124,600, $91,200, $53,800, and $35,900, respectively [22]. Preventing macrosomia/LGA may lead to a reduction in those long-term medical costs.

Besides macrosomia and LGA, other maternal and infant clinical outcomes in the SC group were also less prevalent as compared with the UC group, including preterm birth (18/339 vs. 28/361), Apgar score at 1 min<7 (0/339 vs. 7/361), premature rupture of membrane (49/339 vs. 69/361) and more [14]. We did not include these potential health benefits in our cost-effectiveness analysis as they were not statistically significant between the two groups in our trial, possibly due to inadequate statistical power to detect a difference. Adding these potential health benefits to the possible benefit from improvement in lifestyle for the mothers and their offspring could further improve the cost-effectiveness of the lifestyle intervention in this population.

Previous cost-effectiveness analyses showed that intensive management of mild GDM was cost-effective in other countries and populations. Moss et al. conducted a cost-consequence analysis of the ACHOIS Trial and reported that intensive management of mild GDM was more expensive compared to routine care but cost-effective at $2,186/quality adjusted life years (QALY) [23]. Ohno, et al. used a decision analytic model to compare treating vs. not treating mild GDM in the United States and showed that the cost per QALY was $20,412 [24]. We did not measure health outcomes using QALY, thus we cannot compare our study results directly with results from these previous studies.

Our study has some limitations. First, we did not evaluate the cost-effectiveness of the intervention measured in cost per QALY as we did not collect quality–of-life data due to a limited study budget. If the lifestyle intervention could improve the quality of life for women with GDM, the intervention could be more cost-effective. Second, the direct non-medical and indirect cost estimates might be subject to recall bias, as the costs associated with travelling to follow-up visits, changes in diet, time lost due to outpatient services, and morbidity were collected from a questionnaire at 4–6 weeks after delivery. However, as the main cost items were collected from the Pregnant Women Health Records electronic database records of each participant, the effect of this recall bias would be minimal. Finally, our findings may not be generalizable to other low- and high-income countries due to different clinical diabetes management across countries.

In conclusion, our study was one of the first to evaluate the cost-effectiveness of lifestyle intervention in women with GDM. We found that lifestyle intervention cost less than ¥33,000 for prevention of a macrosomia/LGA infant in China. Considering the potential severe long-term health and economic consequences of GDM and macrosomia/LGA, intensive lifestyle intervention may be an efficient use of health care resources in China, or possibly in other developing countries.

## Supporting information

**S1 Fig. Study flow chart of a lifestyle intervention for women with gestational diabetes in Tianjin, China.**
(DOCX)

**S1 Table. Baseline clinical and biochemical characteristics of study participants, by assignment.** Abbreviations: BMI, body mass index; BP, blood pressure; GDM, gestational diabetes mellitus; GCT, glucose challenge test; HbA1c, hemoglobin A1c; OGTT, oral glucose tolerance test; PG, plasma glucose. † P values were derived from Chi-square Test, Fisher's Exact Test, or Student T Test unless otherwise specified. ‡ Data were reported as median (interquartile range) and P values were derived from Wilcoxon Two-Sample Test.
(DOCX)

**S2 Table. Costs associated with routine surveillance, obstetric and neonatal complications due to gestational diabetes, per study participant, by study arm.** All costs were reported in Chinese yuan (CNY, ¥) (1 CNY: 0.159 USD, 31 December 2012). MI, Maternity insurance. IP, Individual payment.
(DOCX)

## Acknowledgments

The authors would like to express special thanks to the obstetricians and other health professionals in the 64 primary care hospitals and 6 district women and children's health care institutions involved in the trial.

## Author Contributions

**Data curation:** Weiqin Li.

**Formal analysis:** Weiqin Li.

**Funding acquisition:** Huiguang Tian, Xilin Yang.

**Investigation:** Cuiping Zhang, Xilin Yang.

**Methodology:** Ping Zhang, Xilin Yang.

**Project administration:** Junhong Leng, Ping Shao.

**Supervision:** Fuxia Zhang, Ling Dong.

**Validation:** Ping Zhang.

**Writing – original draft:** Weiqin Li.

**Writing – review & editing:** Zhijie Yu, Juliana C. N. Chan, Gang Hu, Xilin Yang.

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
