## [Decision Letter · Decision Letter 0]

19 May 2020

PONE-D-20-12277

Within-trial cost-effectiveness of lifestyle intervention using a 3-tier shared care approach on pregnancy outcomes in Chinese women with gestational diabetes

PLOS ONE

Dear Dr. Li,

Thank you for submitting your manuscript to PLOS ONE. After careful consideration, we feel that it has merit but does not fully meet PLOS ONE’s publication criteria as it currently stands. Therefore, we invite you to submit a revised version of the manuscript hat addresses the points raised during the review process.

We would appreciate receiving your revised manuscript by 07/18/2020. To enhance the reproducibility of your results, we recommend that if applicable you deposit your laboratory protocols in protocols.io, where a protocol can be assigned its own identifier (DOI) such that it can be cited independently in the future. For instructions see: http://journals.plos.org/plosone/s/submission-guidelines#loc-laboratory-protocols

We look forward to receiving your revised manuscript.

Kind regards,

Linglin Xie

Academic Editor

PLOS ONE

Journal Requirements:

2. Thank you for submitting the above manuscript to PLOS ONE. During our internal evaluation of the manuscript, we found significant text overlap between your submission and the following previously published works, some of which you are an author.

https://translational-medicine.biomedcentral.com/articles/10.1186/s12967-014-0290-2

Please revise the manuscript to rephrase the duplicated text, cite your sources, and provide details as to how the current manuscript advances on previous work. Please note that further consideration is dependent on the submission of a manuscript that addresses these concerns about the overlap in text with published work.

The authors have declared that no competing interests exist.

We note that you received funding from a commercial source: Lilly Diabetes

b. Thank you also for including your funding statement; "This project was supported by the National Key Research and Development Program of China (Grants No: 2018YFC1313900, 2018YFC1313903, 2016YFC1300101 and 2016YFC0900602), and BRIDGES (Project No: LT09-227). BRIDGES is an International Diabetes Federation program supported by an educational grant from Lilly Diabetes."

Please include this amended Role of Funder and Competing Interests statement in your cover letter; we will change the online submission form on your behalf.

Reviewers' comments:

Reviewer's Responses to Questions

**Comments to the Author**

1. Is the manuscript technically sound, and do the data support the conclusions?

Reviewer #1: Yes

Reviewer #2: Partly

2. Has the statistical analysis been performed appropriately and rigorously? 

Reviewer #1: Yes

Reviewer #2: Yes

3. Have the authors made all data underlying the findings in their manuscript fully available?

Reviewer #1: Yes

Reviewer #2: Yes

4. Is the manuscript presented in an intelligible fashion and written in standard English?

Reviewer #1: Yes

Reviewer #2: Yes

5. Review Comments to the Author

Reviewer #1: Major comment:

1. Could you provide more details about how you randomly assigned the women into SC or UC group? Or provide some statistics to show no other factors would cause the differences regarding the pregnancy outcomes between two groups.

Minor comment:

1. Some numbers in table 3 are not correct.

2. What are the p-values of : macrosomia [the SC group vs. the UC group: 11.2% (38/339) vs. 17.5% (63/361)] and LGA [the SC group vs. the UC group: 13.0% (44/339) vs. 19.9% (72/361)]

Reviewer #2: Based on the data from one of the biggest city in CHINA, the authors evaluated the cost-effectiveness of lifestyle intervention in women with GDM and found that lifestyle intervention cost less than ¥33,000 for prevention of a macrosomia/LGA infant in China, so intensive lifestyle intervention may be an efficient use of health care resources in China. Although this paper provides potentially interesting new observations, there are some significant issues that need to be addressed for it to be suitable for publication.

Major point:

1. Only 339 from the SC and 361 from UC are availabel finally, its a very important information , should be shown in the abstract.

2. The p value need to be shown in the first paragraph of the results between different groups, without that how to say "the SC significantly reduced the rates of macrosomia....."

Minor point:

3. The unit in the sentence ‘35 kcal/day/kg for one kilogram body weight per day for those

with BMI at <18.5 kg/m2; 30–35 kcal/day for those with BMI at 18.5-23.9 kg/m2;

25-30 kcal/day/kg for those with BMI at 24.0-27.9 kg/m2; 25 kcal/day/kg for those

BMI at 28.0 kg/m2 and more” is not clear.

4. macrosomia should be a special kind of LGA, so the defination of LGA need to exclude macrosomia.

5. the language of this manuscript need to be improved.

6. PLOS authors have the option to publish the peer review history of their article (what does this mean?). If published, this will include your full peer review and any attached files.

Reviewer #1: No

Reviewer #2: No

---

## [Author Response · Author response to Decision Letter 0]

17 Jul 2020

PONE-D-20-12277

Response to Reviewer #1

Reviewer #1: Major comment:

1. Could you provide more details about how you randomly assigned the women into SC or UC group? Or provide some statistics to show no other factors would cause the differences regarding the pregnancy outcomes between two groups.

Response: Thank you for your comment. Details of the randomization were described in our previous publication. Following the suggestion, we now added more details of randomization as follows: “A simple randomization procedure without replacement (i.e., by the time sequence of visits to the clinic and a list of priori computer-generated random assignment status by X.Y.) was used to perform the random assignment by the intervention team members.” We also added a sentence to show no significant differences regarding the pregnancy outcomes between two groups as “There were no significant differences between the two groups for the baseline clinical and biochemical characteristics.”.

Minor comment:

1. Some numbers in table 3 are not correct.

Response: Thank you for the careful reading . We have corrected the error. .

2. What are the p-values of : macrosomia [the SC group vs. the UC group: 11.2% (38/339) vs. 17.5% (63/361)] and LGA [the SC group vs. the UC group: 13.0% (44/339) vs. 19.9% (72/361)]

Response: We have added the p values in the first paragraph of the results.

Response to Reviewer #2

Reviewer #2: Based on the data from one of the biggest city in CHINA, the authors evaluated the cost-effectiveness of lifestyle intervention in women with GDM and found that lifestyle intervention cost less than ¥33,000 for prevention of a macrosomia/LGA infant in China, so intensive lifestyle intervention may be an efficient use of health care resources in China. Although this paper provides potentially interesting new observations, there are some significant issues that need to be addressed for it to be suitable for publication.

Major point:

1. Only 339 from the SC and 361 from UC are availabel finally, its a very important information , should be shown in the abstract.

Response: Thank you for your comment. We have added this information in the abstract.

2. The p value need to be shown in the first paragraph of the results between different groups, without that how to say "the SC significantly reduced the rates of macrosomia....."

Response: Thank you for your comment. We have added the p values in the first paragraph of the results.

Minor point:

3. The unit in the sentence ‘35 kcal/day/kg for one kilogram body weight per day for those with BMI at <18.5 kg/m2; 30–35 kcal/day for those with BMI at 18.5-23.9 kg/m2;

25-30 kcal/day/kg for those with BMI at 24.0-27.9 kg/m2; 25 kcal/day/kg for those

BMI at 28.0 kg/m2 and more” is not clear.

Response: Thank you for your comment. We have revised this sentence.

Now it reads as “35 kcal per day for one kilogram of body weight (35 kcal/day/kg) for women with BMI at <18.5 kg/m2; 30–35 kcal/day/kg for women with BMI at 18.5-23.9 kg/m2; 25-30 kcal/day/kg for women with BMI at 24.0-27.9 kg/m2; 25 kcal/day/kg for women with BMI at 28.0 kg/m2 and more”

4. macrosomia should be a special kind of LGA, so the defination of LGA need to exclude macrosomia.

Response: We thank the reviewer for this insight. We agree that macrosomia is a special kind of LGA. However, in our original RCT design, we used the two outcomes as sperate outcome measures for the trial. We defined LGA as birth weight above the gender- and gestational age- specific 90th percentiles using the Tianjin local references, while macrosomia as birth weight ≥4,000 gram. The two measures were both prior defined primary endpoints for our RCT. We reported the effectiveness of intervention using both endpoints in our previous published main analysis. So we preferred not to exclude macrosomia from LGA to keep consistency with the main report. 

5. the language of this manuscript need to be improved.

Response: Thank you for your comment. Now, a native English editor has copyedited the paper.

---

## [Decision Letter · Decision Letter 1]

3 Aug 2020

Within-trial cost-effectiveness of lifestyle intervention using a 3-tier shared care approach for pregnancy outcomes in Chinese women with gestational diabetes

PONE-D-20-12277R1

Dear Dr. Li

We’re pleased to inform you that your manuscript has been judged scientifically suitable for publication and will be formally accepted for publication once it meets all outstanding technical requirements.

Kind regards,

Linglin Xie

Academic Editor

PLOS ONE

Additional Editor Comments (optional):

Reviewers' comments:

Reviewer's Responses to Questions

**Comments to the Author**

1. If the authors have adequately addressed your comments raised in a previous round of review and you feel that this manuscript is now acceptable for publication, you may indicate that here to bypass the “Comments to the Author” section, enter your conflict of interest statement in the “Confidential to Editor” section, and submit your "Accept" recommendation.

Reviewer #1: All comments have been addressed

Reviewer #2: All comments have been addressed

2. Is the manuscript technically sound, and do the data support the conclusions?

Reviewer #1: Yes

Reviewer #2: Yes

3. Has the statistical analysis been performed appropriately and rigorously? 

Reviewer #1: Yes

Reviewer #2: Yes

4. Have the authors made all data underlying the findings in their manuscript fully available?

Reviewer #1: Yes

Reviewer #2: Yes

5. Is the manuscript presented in an intelligible fashion and written in standard English?

Reviewer #1: Yes

Reviewer #2: Yes

6. Review Comments to the Author

Reviewer #1: (No Response)

Reviewer #2: My commets to author has been responsed in the new version this time, not only the data but also the statistical analysis been performed appropriately, the language also been improved, its valuable to be published.

7. PLOS authors have the option to publish the peer review history of their article (what does this mean?). If published, this will include your full peer review and any attached files.

Reviewer #1: No

Reviewer #2: No

---

## [Editor Report · Acceptance letter]

10 Aug 2020

PONE-D-20-12277R1 

Within-trial cost-effectiveness of lifestyle intervention using a 3-tier shared care approach for pregnancy outcomes in Chinese women with gestational diabetes 

Dear Dr. Li:

I'm pleased to inform you that your manuscript has been deemed suitable for publication in PLOS ONE. Congratulations! Your manuscript is now with our production department. 

Kind regards, 

on behalf of

Dr. Linglin Xie 

Academic Editor

PLOS ONE